# “I Don’t Think It’s on Anyone’s Radar”: The Workforce and System Barriers to Healthcare for Indigenous Women Following a Traumatic Brain Injury Acquired through Violence in Remote Australia

**DOI:** 10.3390/ijerph192214744

**Published:** 2022-11-09

**Authors:** Michelle S. Fitts, Jennifer Cullen, Gail Kingston, Elaine Wills, Karen Soldatic

**Affiliations:** 1Institute for Culture and Society, Western Sydney University, Parramatta, NSW 2751, Australia; 2Menzies School of Health Research, Charles Darwin University, Alice Springs, NT 0871, Australia; 3Australian Institute of Tropical Health and Medicine, James Cook University, Cairns, QLD 4878, Australia; 4Synapse Australia, Brisbane, QLD 3356, Australia; 5College of Healthcare Sciences, James Cook University, Cairns, QLD 4878, Australia; 6Townsville Hospital and Health Service, Townsville, QLD 4814, Australia; 7School of Social Sciences, Western Sydney University, Parramatta, NSW 2751, Australia

**Keywords:** traumatic brain injury, Aboriginal and Torres Strait Islander, women, family violence, Indigenous, remote

## Abstract

Aboriginal and Torres Strait Islander women experience high rates of traumatic brain injury (TBI) as a result of violence. While healthcare access is critical for women who have experienced a TBI as it can support pre-screening, comprehensive diagnostic assessment, and referral pathways, little is known about the barriers for Aboriginal and Torres Strait Islander women in remote areas to access healthcare. To address this gap, this study focuses on the workforce barriers in one remote region in Australia. Semi-structured interviews and focus groups were conducted with 38 professionals from various sectors including health, crisis accommodation and support, disability, family violence, and legal services. Interviews and focus groups were audiotaped and transcribed verbatim and were analysed using thematic analysis. The results highlighted various workforce barriers that affected pre-screening and diagnostic assessment including limited access to specialist neuropsychology services and stable remote primary healthcare professionals with remote expertise. There were also low levels of TBI training and knowledge among community-based professionals. The addition of pre-screening questions together with professional training on TBI may improve how remote service systems respond to women with potential TBI. Further research to understand the perspectives of Aboriginal and Torres Strait Islander women living with TBI is needed.

## 1. Introduction

Addressing violence against women and children is an international priority [1,2]. Violence incorporates psychological abuse, as well as physical and sexual abuse, in addition to various forms of coercion and control of women’s everyday behaviours [3]. First Nations women in Australia and other settler colonial countries such as the United States of America experience some of the highest rates of physical violence in the world [4,5,6]. Compared with other Australian women, Aboriginal and Torres Strait Islander women are 32 times more likely to be hospitalised due to family-related assaults [4] and are five times more likely to be fatally injured as a result of intimate partner violence [7]. Physical violence can result in injuries that affect the functioning of the brain such as traumatic brain injury (TBI) [8]. Classified under the umbrella term of acquired brain injury (ABI), TBI is defined as damage to, or alteration of, brain function due to a blow or force to the head; it can be classified as mild (also referred to as concussion), moderate or severe [9]. Violence is a leading cause of TBI for First Nations women internationally, including Aboriginal and Torres Strait Islander women [10,11]. There is also growing recognition that non-fatal strangulation should be included under the umbrella of TBI, particularly in the context of violence. The rate of head injury due to assault (1999–2005) among Aboriginal and Torres Strait Islander women is 69 times the rate of other Australian women [12]. This is likely to be an underestimation due to factors affecting reporting violence including stigmatisation attached to experiencing family violence, fear of retaliation from the perpetrator, and the violence often experienced through racialised policing and criminalisation when Aboriginal and Torres Strait Islander peoples report and seek protection from police [13]. Despite this strong association, the nexus of violence and TBI has been largely overlooked in research, practice, and policy arenas, with only a small number of TBI studies completed with Aboriginal and Torres Strait Islander peoples [1,14,15,16]. None of these studies specifically focus on the experiences of Aboriginal and Torres Strait Islander women only.

### 1.1. Background

Access to the healthcare system is critical for women who experience TBI as a result of family violence and is recognised as an important step towards effectively addressing violence against women [17]. Early identification of TBI through comprehensive pre-screening, including questions regarding the potential to have experienced a TBI (e.g., number of times consciousness has been lost following a blow to the head; occurrence of strangulation) [18,19] and screening, as well as timely neurorehabilitation interventions, are an essential component of the healthcare system, when completed appropriately, as they have the potential to mitigate the impact of brain injury and can trigger planning for personalised supports. Access to allied healthcare and specialist supports (e.g., neuropsychology services) is required so women can be assessed accurately and, in combination with their general practitioner, be referred to necessary support to understand the nature of their injury and their needs for rehabilitation, recovery, and adjusting to the impacts on their everyday lives [18,20]. Specialists can also provide women with medical certificates and reports required to access the National Disability Insurance Scheme and additional financial support such as necessary income support payments that are specifically designed for persons with disabilities [21]. Failure to identify TBI can lead to symptoms being misattributed to reasons other than the injury and result in deprivation of much-needed services.

Access to healthcare and specialists remains limited for many Aboriginal and Torres Strait Islander peoples in remote areas in Australia [22,23,24]. Obtaining equitable access to healthcare is largely associated with persisting issues of workforce undersupply and geographical maldistribution, retention of the health workforce, and funding [22,23,24]. An Australian study found that 20% of nurses and Aboriginal Health Practitioners remained working at the same remote primary healthcare clinic 12 months after commencing and that half left within four months [23]. According to the Australian Institute of Health and Welfare, in remote areas, the psychologist full-time equivalent rate per 100,000 population is nearly one-quarter of that of major cities (29 versus 104) [25]. It is assumed that there are similar distributions for neuropsychologists, considering the relatively low numbers practising nationally (around 5% of all psychologists) [26]. TBI studies conducted with service providers in other similar settler nations described “massive chasms” within systems that prevent First Nations women from timely TBI pre-screening and diagnosis [27]. Lack of adequate community infrastructure, low levels of access to primary healthcare, community health centres and hospitals, and the costs associated with neuropsychologist assessments were some of the major barriers for First Nations women accessing healthcare and specialist care [27]. Resource barriers can be coupled with racist and prejudiced assumptions made by staff about First Nations peoples who access medical healthcare for a TBI, which can in turn create further barriers to access and use of service systems [15,27]. 

### 1.2. Aim

The study reported here is part of a three-year project to document and understand the needs and priorities of Aboriginal and Torres Strait Islander women who have acquired a TBI through family violence (funded by the Australian Research Council, #210100639) [28]. The larger project has the following participant cohorts: (1) Aboriginal and Torres Strait Islander women (aged 18+) with lived experience of head injuries or diagnosed TBI as an outcome of family violence, (2) family members and carers of women with lived experience of head injuries or diagnosed TBI as an outcome of family violence, (3) hospital staff, and, (4) service providers including family violence, health, housing and crisis accommodation, social, disability, mental health, and legal sectors, as well as local community groups. This study aims to identify community-based service provider perspectives on workforce and system-related barriers affecting pre-screening and healthcare access for Aboriginal and Torres Strait Islander women after experiencing a potential TBI following family violence. The focus of the study on service providers is deliberate, as culturally appropriate and safe servicing of Aboriginal and Torres Strait Islander women with a potential TBI as an outcome of violence is vital to ensure the necessary service and workforce processes, supports, and skills are available.

## 2. Materials and Methods

### 2.1. Setting 

Service providers who participated in this study were all located in a remote region in the Northern Territory, Australia. The Northern Territory population of 228,600 (2016 census) is dispersed over 1.4 million square km [28]. Within this location, around 25% of the region’s population self-identify as Aboriginal and/or Torres Strait Islander individuals [28] and have ongoing connections to country, kinship lore, and family. Within the study region, there are many local Aboriginal languages that are socially, culturally, and linguistically distinct. These are living languages, and they are the primary languages in use among families and kinship networks within the region. Many of the service providers who work or are based in the main township in the region also provide services to women in remote communities, covering large distances by road or air. The exact location of the study region is not revealed nor are descriptors for the quotes provided. This is to protect the confidentiality of the community and service providers.

### 2.2. Participants and Recruitment

A qualitative study incorporating semi-structured interviews and focus groups with service professionals was conducted between January 2022 and July 2022. Using purposeful sampling, the research team identified and approached a variety of different stakeholder groups that were regarded as highly knowledgeable and working closely with Aboriginal and Torres Strait Islander women who had experienced or were experiencing family violence [29]. Professionals represented acute and crisis services to long-term programs and services, within disability, family violence, health, housing and accommodation, social, mental health, and justice and correctional sectors. Their core business to support women often varied and could involve advocacy, individual support, education, and delivery of programs. In order to achieve saturation, purposeful sampling was coupled with a snowball approach to recruit participants, whereby participants were able to recommend other relevant agencies in the region for the research team to approach [30]. 

### 2.3. Traumatic Brain Injury Knowledge Workshops

Prior to data collection, culturally safe workshops were delivered (in person or online) by two Aboriginal and Torres Strait Islander educators via a partner organisation of the project. The workshops were intended to operate as a two-way learning opportunity to improve the knowledge base of service professionals, including to improve their terminology and language to articulate the experiences of their clients with acquired head injury and increase the quality of the data collected from interviews and focus groups [31]. The partner organisation that conducted the workshops is a non-government organisation dedicated to reconnecting the lives of people who are affected by brain injury and to building partnerships with Aboriginal and Torres Strait Islander peoples in order to incorporate and support their ideas, strengths, and leadership. The education package was a pre-existing resource the partner organisation had developed with and for Aboriginal and Torres Strait communities and service providers working in regional and remote locations. Topics covered include the common psychological, behavioural, and cognitive changes that can occur with acquired brain injury and how other conditions can display similar symptoms (e.g., mental health conditions). The facilitators drew upon real case examples from work the partner organisation does with Aboriginal and Torres Strait Islander peoples with acquired brain injuries in communities, as well in the court and prison systems. Augmenting the workshops were digital storytelling resources of people sharing their personal journeys following traumatic brain injury, developed under a previous TBI project [32]. 

### 2.4. Data Collection 

Interviews and focus groups were conducted by a non-Indigenous researcher (M.F.) and an Aboriginal researcher (E.W.). The majority of participants completed an interview or focus group, with a small number of participants completing both an interview and a focus group. Twenty-three interviews and three focus groups were conducted face to face with a further single interview and focus group conducted through video conference to comply with COVID-19 restrictions. A total of 38 participants were recruited to the study. Interviews ranged 45 to 101 min. The four focus groups ranged from 45 to 73 min in duration with an average of 58 min. A topic guide was created for the semi-structured interviews and focus groups by the research team. The topic guide covered questions about types of supports and services available to women, enablers and barriers women experience when accessing services, challenges for service providers in supporting women, suggestions for reform, and improvements for policy and service delivery. Questions in the interview schedule included: What are the main reasons women seek help from you? Can you tell me about your experiences helping women? (What works well? What does not work so well?) What are specific barriers and issues for women in accessing support? The questions asked could vary depending on participant expertise. Participants were instructed that the interview or focus group would explore general principles and concepts of TBI and family violence rather than individual stories to encourage higher level concepts for analysis. An iterative approach was taken by adding probing questions as the interviews and focus groups progressed, on the basis of participant stories. 

### 2.5. Data Analysis

A professional audio typist transcribed all audio recordings, and these were checked for accuracy. All written notes were transcribed. Transcripts, fieldnotes, and observations were managed with NVivo software [33]. Inductive thematic analysis was used as there is currently limited evidence from the perspectives of community-based service providers that support Aboriginal and Torres Strait Islander women. This approach enabled the emergence of themes from patterns of meaning and revealed unexpected perceptions [34]. Thematic analysis was conducted by the first author. To strengthen the validity of the findings, the transcripts were read several times to increase familiarity with the data and facilitate in-depth examination to uncover meanings and relationships among the emerging codes. A coding framework was collaboratively developed to determine a coding structure, with feedback provided by all co-authors. Coding involved familiarisation with the data, and then generation of descriptive codes before interpretive coding took place [35]. Established guidelines for qualitative studies were used to enhance the quality of the research. The broad patterns that appeared across the interviews and focus groups including existing target topics and unanticipated issues were identified and labelled as themes. Triangulation, a process of establishing validity, was achieved through the following steps: (i) ensuring that the identified themes were supported by findings in the literature on other TBI and family violence, to provide a credibility check, and (ii) presenting findings to the project advisory board and some of the participants to ensure that they resonated with others in similar positions and with similar experiences [36]. 

### 2.6. Ethics Approval 

This study was approved by the Central Australian Human Research Ethics Committee (CA-21-4160) and Western Sydney University Human Research Ethics Committee (H14646). The study also received approval from Aboriginal community-controlled, legal, and family violence research committees and boards. Australia’s National Health and Medical Research Council protocols for research with Aboriginal and Torres Strait Islander peoples were followed at each step of the project. All participation was voluntary, and all participants signed consent forms.

## 3. Results 

According to service providers, there were several workforce and system barriers that impacted upon access to healthcare to Aboriginal and Torres Strait Islander women who had experienced a TBI due to family violence. Four themes were discerned from the data: (1) qualities of the remote primary healthcare workforce, (2) limited pathways to healthcare access for women with mild head injuries, (3) lack of specialist care servicing the region, and (4) workforce knowledge and pre-screening of traumatic brain injury—tools that could secure optimal outcomes.

### 3.1. Qualities of the Remote Primary Healthcare Workforce

The remote primary healthcare workforce outside of the main township was described by service provider participants as having a “high turnover”. According to staff, turnover of the primary health workforce contributed to inconsistent healthcare access and affected the quality of the relationship between health staff and community members who use remote healthcare clinics. Healthcare clinics that provide services to remote communities usually consist of small teams including remote area nurses and Aboriginal and Torres Strait Islander health practitioners with additional services provided by visiting general practitioners, and allied health professionals. Remote areas nurses were often termed “fly-in, fly-out” staff [22,23], with staff frequently mentioning new staff working in remote clinics did not have experience working in or familiarity with a remote context:

*The nurses in those communities, there might be nurses in there who haven’t worked in those [communities], they fly in, fly out, they could be new nurses who miss the signs [of traumatic brain injury], because they’re not as used to it*.(Service provider 2)

In combination with the staff turnover within the primary healthcare workforce, service provider participants felt the high workloads of health professionals in remote primary health clinics could also be a factor for a potential TBI being overlooked when women present to the clinic: 

*They do the best down there, the nurses, in the clinics, so I don’t want to make it out that it’s horrendous what they’re doing out there. They’re overrun, they’re understaffed. So, it’s not surprising that things get missed*.(Service provider 2)

The workforce shortages also had an impact on clinic access reliability for women in some remote communities including ad hoc opening hours, best captured by service provider 1: *And there might not even be a clinic for her to go to.* Participants provided examples of clients who were required to access healthcare in other communities. To respond to these circumstances, women were reliant on accessing a working vehicle in order to travel hundreds of kilometres to a neighbouring community to access healthcare. For most women, attempting to drive vast distances after experiencing family violence would also generate further safety risks, especially given the distances between towns across the largely unserviced road network of the area.

When speaking about the high turnover of the remote primary healthcare workforce, staff provided examples of how this affected the information their clients would disclose when they presented to the clinic following an assault. One service provider participant provided a case of a client who had disclosed an assault to a staff member at their service because of the long-term relationship and trust between the client and staff member. As the participant explained, the client did not disclose this information when she first presented to the clinic: 

*It was only through our own questioning of her, careful questioning of her what happened, and our relationship, that she told us the severity of the assaults, but the clinic wasn’t aware of it*.(Service provider 3)

### 3.2. Limited Pathways to Healthcare Access for Women with Mild Head Injuries 

Interviewed staff working in the main town and across the region thought the remote healthcare pathways and systems were not designed to respond to all severity levels of TBI across the region. The medical emergency retrieval service operating in the region was viewed as critical for women in remote communities outside of the town to access more comprehensive emergency care. The retrieval service consisting of acute specialist care staff could support primary healthcare staff in the clinics servicing remote communities to determine if a patient needed to be evacuated from the community. If it was decided that a patient needed more critical care or was at further risk of harm, the medical emergency retrieval service organised and coordinated aeromedical or road evacuation and transportation to the major hospital in the region. In the narratives, staff discussed their clients who had sustained head injuries and their ability to access healthcare outside of their remote community. Staff provided examples where women presented to their local primary healthcare clinic following a head injury, with the ability to walk and communicate with primary healthcare staff. These clients were not evacuated out of the community, with their condition and symptoms considered “manageable”. For clients who were not transported out of the community, staff raised concerns about the gaps in medical investigation (e.g., scans) and evidence in their records as explained by the following participants:

*Women are often not evac-ed out following a head injury, if it’s assessed to not be an urgent thing, so might not necessarily be getting CT [computerised tomography] scans and things like that.*.(Service provider 7)

*She would have needed to go to [the major town] to get an MRI [magnetic resonance imaging] and they didn’t deem it as, I don’t know what the wording was, but was [not] significant enough to fly her all the way to [major town] for the MRI. So, she never got it.*.(Service provider 9)

For women who were not transported out of the community, service providers were unaware of any specific follow-up support post-injury provided to their clients to determine if they were experiencing ongoing symptoms related to their head injury. 

### 3.3. Lack of Specialist Care Servicing the Region 

A common issue raised by staff was the lack of diagnosed TBI: *It’s like, okay, well something’s going on here. We very rarely will have someone with a diagnosis, if ever, to be honest.* (Service provider 4). Staff with long-term work experience within the region (10+ years) explained longstanding difficulties of connecting their clients with neuropsychologists due to an undersupply of specialist services. Participants felt that the number of resident and visiting specialists (such as neuropsychologists) working in the region was not reflective of the need. The absence of specialist services created significant challenges for staff obtaining medico-legal assessments and reports, which was often a requirement for a range of client services and supports. Specialist reports were also critical evidence in the court system. As one participant explained:

*You get their medical histories and their police records, and they’ve just been basically pummelled within an inch of their life for all of their life in terms of re-occurring head injuries, re-occurring stabbings and assaults and you just look at the totality of it and go, how does this person even function at any level? But of course the problem is often being able to, there are really very few people who are specialised to give an opinion, such as [a] neuropsychologist, to come up here or work in this region so it’s very hard to get a proper diagnosis. Without a diagnosis you then have problems in terms of convincing a court that there is a disability at play here which is an important factor to be taken into account when sentencing them or when a determination is made.*.(Service provider 23)

Lack of a formal diagnosis created difficulties for staff and their service. Family violence, crisis accommodation, and health service sector staff, in particular, described it as “challenging” to incorporate TBI adequately into their case management, as well as develop appropriate response plans to meet the needs of their clients when such clients displayed signs of a TBI. As one participant explained, even if the common behavioural and psychological consequences of TBI were exhibited by their clients there were no formal assessment reports they could draw upon to inform their own case management plans: 

*I guess it’s difficult, because do we assume that all women have a head injury? Like none of the women that we have, I don’t know any who actually have a confirmed diagnosis of a TBI, like that would be the issue. So that’s the difficulty for us as well, because they haven’t actually been formally diagnosed with anything*.(Service provider 3)

In attempts to address these gaps of medical investigation, some legal service providers paid for specialist assessments and reports. This was usually once women were in contact with the criminal justice system. As services were not fully funded to cover costs for specialist evaluations, this option was not afforded to all clients. 

### 3.4. Workforce Knowledge and Pre-Screening of Traumatic Brain Injury—Tools That Could Secure Optimal Outcomes

Community-based service providers were considered a crucial gateway for women to be linked with the right health and medical services. Some of those who participated in this study had worked with clients for several years. Family violence training, trauma informed practice training, and education on other forms of ABI including foetal alcohol spectrum disorder (FASD) were the most common programs and courses interviewed frontline staff had completed. None of the interviewed staff had completed TBI education, as the following participant explained:

*We have training about a lot of different things, but training around how to help people with an acquired or a traumatic brain injury is not something I’ve come across in this space, in this case management space, [in contrast to] trauma informed care person, [and] client-centred practice, that sort of training. We get ADD [attention-deficit disorder] workshops, we get domestic and family violence workshops, disability support workshops, but nothing around brain injury.*.(Service provider 26)

As described in the methods section, TBI workshops were delivered to service providers prior to interviews and focus groups. These workshops enabled participants to reflect upon their own practice and the assumptions they or their colleagues had made about some clients’ behaviour and symptoms. As one participant explained, staff have associated these symptoms with mental health conditions and long-term alcohol misuse, with TBI not considered:

*As a service provider, you think, why did I not think about this earlier? Now that we are having this conversation, I can think of a few clients, one in particular who we thought had mental health issues and also had a long history of AOD [alcohol and other drugs], but it is likely that she was experiencing a brain injury after the years of violence she had suffered. But we didn’t pick that up when we were working with her*.(Service provider 16)

Participant responses indicated the need for more general awareness and education of the nexus of violence and TBI at the workforce level across sectors to better enable support for clients. Some participants suggested training could provide guidance for staff on how to identify potential signs of TBI and to ask and discuss TBI with clients. As one participant stated, TBI education would also facilitate a deeper understanding for staff of how their clients experience and live with TBI:

*This really is an area that, I think, there’s just maturity and skills to be built for us and to be able to believe, just for us to recognise what’s happening for them. Because, I think, everyone that works here believes [it is] happening and it’s very real for some women, where to go in terms of the support that we can provide. I think, it’s really good to do this type of training. Because also with the barriers, that in itself is a barrier for women, right, because they’re not, the service providers aren’t fully understanding what’s happening to them*.(Service provider 10)

In addition to building knowledge and skill capabilities, pre-screening [18] for TBI was identified as a gap. Most service sectors involved in the study routinely screen clients for family violence as part of their risk assessment process. Participants described the benefits of screening for family violence including helping to identify those at risk and enabling early intervention. While these tools are highly valued by staff, it was acknowledged that there were no specific questions to collect information from clients about potential TBI they may have experienced as an outcome of family violence: *We’ll screen for domestic violence, but we don’t screen for specific injuries*. (Service provider 4). Of the service providers who did screen for TBI, it was mainly legal services who participated in the study that asked new clients at intake about TBI. Staff from two service providers who did not formally screen for TBI provided examples where they added detailed information about “suspected head injury” in case notes and client management plans. These plans were used to connect their clients to support services. Staff spoke about the need for inclusion of TBI in family violence-related screening tools, as well as training for case workers about how to respond to information they collect from asking about potential TBI including actioning referrals to appropriate services: 

*But if there was a screening tool for domestic violence specialist services that was looking at brain injury in a culturally appropriate way for our way. Something like that, that would support workers to know and ask the questions and then also, support workers to know what to do*.(Service provider 8)

## 4. Discussion

The study illustrated that there are a range of workforce barriers that prevent pre-screening and identification of a potential TBI, as well as access to healthcare. Healthcare systems involved in responding to TBI (e.g., emergency and health systems) are attuned to this type of injury as a “one-time event”, where a survivor who has sustained a moderate to severe TBI accesses hospital care and follows a pathway of recovery and rehabilitation [15,37,38]. Dissimilar to other mechanisms of TBI such as falls and motor vehicle crashes, injuries to the head in the context of violence are usually frequent and repetitive [39]. From the narratives, mild TBIs are managed in the community. The remote healthcare services are further challenged by the well-documented, longstanding issues associated with healthcare provision in remote areas in Australia and elsewhere including workforce turnover [40,41,42,43], combined with few neuropsychologist services. None of the service providers involved with the project had completed TBI education, and this is likely to be a shared experience across service sectors who work closely with women who experience violence as has been identified in other settler colonial countries such as Canada [27,44]. Lastly, few community-based service providers routinely pre-screen for potential TBI [18,45]. Addressing these gaps is critical to the longer-term health of women as healthcare and medical documentation are a requirement of critical economic supports, such as the Disability Support Pension, particularly for women experiencing housing instability and living in high-risk environments [21,46].

### 4.1. Implications for Policy and Practice

The findings suggest several practice implications for remote service providers in Australia. First, increasing education and awareness within the workforce would assist services in responding to women who are cognitively, behaviourally, and psychologically impacted by TBI and to link them with early intervention for support and recovery. For remote health professionals, this would include greater investment in resources to prepare and orient new health professionals regarding family violence and TBI. More broadly, building the knowledge and skill set of the workforce across key frontline service sectors may help service workers to be more adept in asking pre-screening questions and enhance their confidence about how to respond if women report experiencing a potential TBI [18,44,47]. Examples include awareness that women may not be forthcoming in reporting violence related TBI without being asked, building the knowledge of the common signs of TBI, including that mild TBI can have few overt symptoms, and that TBI symptoms can resemble the effects of alcohol and other drugs, as well as other conditions, and how to support a client post-injury [31,39,48]. 

Second, there should be clear processes for community-based service providers to discuss potential TBI with their clients combined with clear referral pathways to direct their clients to appropriate health professionals and supports. The annual health check for Aboriginal and Torres Strait Islander peoples is a health screen completed to ensure the primary health care received matches patient needs, by encouraging early detection, diagnosis and intervention [49]. Integration of TBI pre-screening into the annual health check as well as in existing violence-related risk assessments may support better identification of a potential TBI [18]. Moreover, culturally appropriate undiagnostic screening tools could also help to assist in the identification of cognitive impairment and complex disability while overcoming the lack of neuropsychologists working in remote Australia [50]. Pre-screening and screening for TBI does need to be conducted in culturally safe and appropriate manner, with consent and communication (e.g., access to an interpreter/translator) needing to be considered in the process [51]. 

Third, this study found that women who have acquired a TBI through violence may not access healthcare immediately following their injury, thus placing emphasis upon the importance of strong and responsive primary healthcare services. Mild TBI symptoms have been shown to last six months [52] making easy access to healthcare combined with meaningful continuity of care and long-term follow-up with a trusted healthcare professional [53] critical to ensuring women are appropriately supported post their injury. Remote health service delivery brings many unique challenges not seen in urban health services related to cost, distance, highly dispersed populations, and access to profession-specific expertise. Strengthening of effective workforce recruitment and retention strategies, such as Aboriginal workforce strategies and rural training pathways, is important to build sustainable remote health workforces. Changing primary healthcare funding to needs-based population funding would also be beneficial to improve health funding equity by building sustainable primary healthcare service delivery in remote Aboriginal and Torres Strait Islander communities, which is an issue across wider Australia. Moreover, there is growing evidence of the benefits of comprehensive interdisciplinary rehabilitation for mild TBI, with integrated therapeutic interventions targeting cognitive, emotional, and physical domains. This type of model has shown to have positive effects on TBI symptoms, general psychological function, depression, participation, and quality of life [54]. Further exploration of this type of model of care for Aboriginal and Torres Strait Islander women in a remote context is recommended. 

Finally, the potential promise of digital health as an alternative model of service has been highlighted by increased use in remote areas during the COVID-19 pandemic to maintain delivery of healthcare. Use of digital health for health screening, education care in mental health, and chronic health conditions is perceived to be feasible, satisfactory, and accessible [55], but there is a limited evidence base for digital health use with Aboriginal and Torres Strait Islander peoples [56]. Further exploration on how digital health may be able to improve service accessibility and its suitability for use with women who have experienced violence-related TBI is required. 

### 4.2. Limitations 

First, the design of this study is not intended to arrive at results that are generalisable. Rather, the goal of this type of inquiry is to develop better, more nuanced understanding of a complex phenomenon and to identify new knowledge that could be transferable to other contexts. As such, the findings should not be read as relevant to all remote locations but should be considered for their transferability to particular sites. Nevertheless, using purposeful sampling arguably gave considerable strength to this study by including a range of diverse and broad perspectives and experience, increasing the potential for transferability of the findings and recommendation for service delivery to Aboriginal and Torres Strait Islander women experiencing family violence. As this study is based on community-based service providers, further research to prioritise the voices of Aboriginal and Torres Strait Islander women with lived experience of TBI connected to family violence to obtain a better understanding of their experiences of these systems is required. Lastly, at the time interviews were being conducted, COVID-19 pandemic measures were in place to minimise risk of transmission including the wearing of masks, with one interview and one focus group completed online. 

## 5. Conclusions

In Australia, the nexus of family violence and TBI and its implications are now being recognised in the formulation of family violence policies and frameworks. Yet, little qualitative research exists with service providers who work with women, particularly Aboriginal and Torres Strait Islander women in remote communities, to activate these policies and frameworks effectively. This study is an important step towards understanding the workforce and system-related barriers for Aboriginal and Torres Strait Islander women accessing medical and healthcare services including pre-screening and assessment following a potential TBI as an outcome of family violence. The findings reinforce the importance of addressing the well-known, existing barriers that Aboriginal and Torres Strait Islander peoples in remote Australia experience when attempting to access primary and specialist healthcare including turnover of staff in primary healthcare and undersupply of specialists. To strengthen a coordinated, integrated response, increasing the low levels of training and education around TBI is critical towards ensuring that women are provided with the right support and referrals.

## Data Availability

Not applicable.

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
