# Peer review of "“I Don’t Think It’s on Anyone’s Radar”: The Workforce and System Barriers to Healthcare for Indigenous Women Following a Traumatic Brain Injury Acquired through Violence in Remote Australia"

_ijerph, 2022, doi:10.3390/ijerph192214744_

Round 1

Reviewer 1 Report

Dear Authors,

Thank you for your submission, I believe it makes a substantial contribution to the body of evidence on access to healthcare for Aboriginal and Torres Strait Islander peoples. I only have one comment, and that is whether there was consideration of interviewing the consumers of services, rather than only those that provide the services. I read that this project is part of a three year study and perhaps the other elements of the study will involve data collection from people with lived experience of domestic and family violence - apologies if this has been stated in the content and I have missed it. 

Best wishes

Reviewer 2 Report

TBI following domestic violence is a highly under-researched area - yet the high prevalence of violence and high risk of injuries makes this a very clinically important area. I was pleased to read this article, particularly given its focus on indigenous women who are under-served.

The article is generally well-written and explained. I have some comments and suggestions below;

Overall comments

1) I am not sure why the authors refer to pre-screening throughout. A pre-screen would generally refer to a screen before an injury had been sustained e.g. a baseline measure. It might be clearer just to refer to a 'screening tool'.

2) The authors interchange between discussion groups and focus groups. I would suggest sticking with focus groups throughout would make it flow better and be more specific for the reader.

3) Neuropsychology services can be helpful following concussion but are not necessarily needed by everyone following a concussion. So best these are used for those where it is most needed, particulalry given the shortages as noted by the authors. A meta-analysis has shown that multidiscisplinary concussion services can help to improve recovery - I would suggest the authors broaden out their recommendations that people need access to rehabilitation following brain injury rather than specifically highlighting neuropsychologists. Given everyone responds to brain injury differently some may need a physiotherapist or an OT more than a neuropsychologist, whereas others will need a neuropsychologist. I think highlighting access to more general supports would be more useful throughout.

4) I found the term 'service provider staff' difficult to read - could this just refer to staff or health professionals?

5) cost didn't come up as an issue for access yet this has been highlighted in previous research. It would be useful to discuss why this was the case here.

6) Existing screening tools such as the HELPS tool need to be referred to.

7) I think it would be useful to add in the discussion about how the needs of indigenous women in Australia might be different given the findings to previous studies and other populations in accessing support after brain injury.

Specific comments

1) Lines 3 and 4 of the abstract, the word support is used twice in the same sentence.

2) The reference [9] given for defining TBI is not what I was expecting I would refer to world health organisation definition

3) I was not clear  how section 2.3 was relevant to this study?

4) in the data collection section I would suggest moving the duration of the interview durations to where it is specified for the focus groups so people can compare the two more easily and to improve flow of this paragraph. I also note in this paragraph that it states that the questions could vary dependent on participant expertise, however no interview schedule is presented. This needs to be included so the reader can see how data was generated by the questions that were asked.

5) Given the focus on indigenous women's experiences I think there needs to be justification for using a western qualitative approach

6) more details are needed on how and who coded the data e.g. two people independently or two taking different transcripts each. How were codes recorded, revised and decided upon?

7) whilst I appreciate the need to protect the privacy of participants. It would be an important context to have a high level of professions of participants e.g. social workers, nurses etc... in order for the reader to see which perspectives are reflected in the study.
